# The Effect of Taper and Apical Diameter on the Cyclic Fatigue Resistance of Rotary Endodontic Files Using an Experimental Electronic Device

Vicente Faus-Llácer [1], Nirmine Hamoud Kharrat [1], Celia Ruiz-Sánchez [1], Ignacio Faus-Matoses [1], Álvaro Zubizarreta-Macho [2,*] and Vicente Faus-Matoses [1]

[1] Department of Stomatology, Faculty of Medicine and Dentistry, University of Valencia, 46010 Valencia, Spain; fausvj@uv.es (V.F.-L.); nirhak@alumni.uv.es (N.H.K.); celia.ruiz@uv.es (C.R.-S.); ignacio.faus@uv.es (I.F.-M.); vicente.faus@uv.es (V.F.-M.)

[2] Department of Endodontics, Faculty of Health Sciences, Alfonso X El Sabio University, 28691 Madrid, Spain

* Correspondence: amacho@uax.es

**Abstract:** The aim of this study was to analyze the effect of the taper and apical diameter of nickel–titanium (NiTi) endodontic rotary files on the dynamic cyclic fatigue resistance. A total of 50 unused conventional NiTi wire alloy endodontic rotary instruments were used in this study. All NiTi endodontic rotary files were submitted to a custom-made dynamic cyclic fatigue device until fracture occurred. The time to failure, the number of cycles to failure, the number of pecking movements, and the length of the fractured file tip were analyzed using the analysis of variance (ANOVA) test. In addition, the Weibull characteristic strength and Weibull modulus were also calculated. The paired *t*-test revealed statistically significant differences between the time to failure, number of cycles to failure, and number of cycles of in-and-out movement of both the apical diameter ($p < 0.001$) and the taper ($p < 0.001$) of NiTi endodontic rotary files; however, the results did not show statistically significant differences between the mean length of the fractured files regarding the apical diameter ($p = 0.344$) and taper study groups ($p = 0.344$). Increased apical diameter and taper of NiTi endodontic rotary files decreased their dynamic resistance to cyclic fatigue.

**Keywords:** endodontics; cyclic fatigue; taper; apical diameter; rotary movement; endodontic rotary files



## 1. Introduction

The introduction of nickel–titanium (NiTi) alloy in endodontic rotary instruments had a great impact on endodontics due to their combination of speed, quality, accuracy, and risk reduction [1]. However, the main drawback of the NiTi endodontic rotary instruments is the fracture of the endodontic files inside the root canal system during the shaping procedures, which prevents root canal system disinfection beyond the fractured instrument and can influence the prognosis of the root canal treatment [2,3].

The fracture of NiTi endodontic rotary files can occur as a consequence of excessive torsion or flexural fatigue [4,5]; the latter being the most frequent cause [6]. Flexural fatigue or cyclic fatigue is caused by the alternating compression and traction cycles that NiTi endodontic rotary files experience at the point of maximum curvature of the root canal system [7]. There are factors related to the cyclic fatigue resistance of NiTi endodontic rotary files: cross-section design [8], the diameter of the inner and outer core [2], the operating speed and torque [9,10], radius and angle of curvature [11], operator capability [12,13], anatomical configuration of the root canal system [14], irrigation solutions [15], sterilization cycles [16,17], and NiTi alloy [3,18].

However, the influence of the taper and apical diameter of NiTi endodontic rotary files on the cyclic fatigue resistance has never been reported, although research has highlighted

that the fracture incidence of NiTi endodontic rotary files increased in curved root canal systems, possibly related to the apical diameter [19] and taper of the NiTi endodontic rotary files [20], mainly at the point of the maximum curvature. This is because large file tapers accumulate a greater amount of internal stress during stress-compression cycles when flexed to accommodate the curvature of the root canal system [21]. However, an increase in the diameter of the NiTi endodontic rotary files can contribute to increasing the resistance to torsional fracture [5].

In addition, many cyclic fatigue testing devices have been proposed in the literature; however, they differ mainly in the ability to reproduce the pecking movements of the operator statically or dynamically during the root canal shaping. Dynamic cyclic fatigue testing devices are recommended because they faithfully reproduce the operator pecking movements; moreover, Loios et al. reported that the pecking movements performed by dynamic cyclic fatigue testing devices extended the fatigue lifetime of the NiTi endodontic rotary files, compared to static cyclic fatigue testing devices [22]. Therefore, a dynamic cyclic fatigue testing device was used in this study.

The aim of this study was to analyze and compare the effect of the taper and apical diameter on the dynamic cyclic fatigue resistance of NiTi endodontic rotary files, with a null hypothesis (H0) stating that the taper and apical diameter would not affect the resistance of NiTi endodontic rotary files in dynamic cyclic fatigue.

## 2. Materials and Methods

### 2.1. Study Design

In this in vitro study, we utilized 50 unused conventional NiTi wire alloy endodontic rotary instruments (RaCe®, La Chaux-De-Fonds, Switzerland), 25 mm in length with a triangular cross-section. All NiTi endodontic rotary files were analyzed using scanning electron microscopy (SEM) (HITACHI S-4800, Fukuoka, Japan) in the Department of Mechanical, Energetic, and Materials Engineering of the School of Industrial Engineering of the University of Extremadura (Badajoz, Spain) under the following exposure parameters: acceleration voltage: 20 kV, magnification from $100\times$ to $6500\times$, and a resolution between $-1.0$ nm at 15 kV and 2.0 nm at 1 kV, to confirm the taper (Figure 1) and apical diameter (Figure 2) values of the previously selected NiTi endodontic rotary files (RaCe®, La Chaux-De-Fonds, Switzerland).

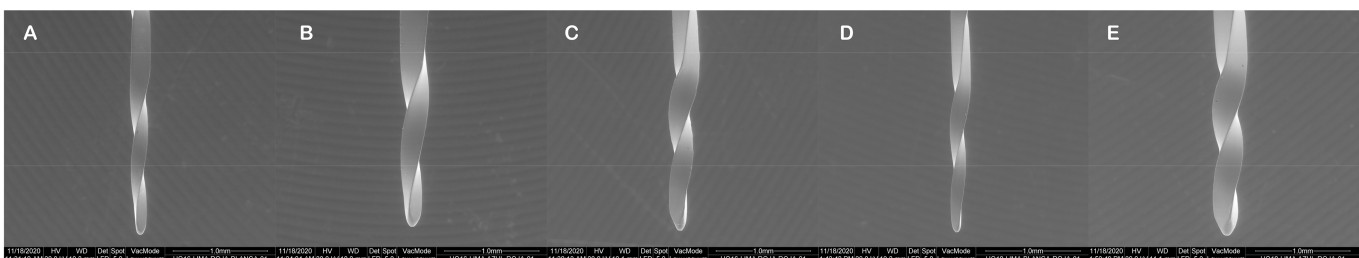

**Figure 1.** Scanning electron microscopy (SEM) analysis of the taper of (**A**) 25.02, (**B**) 25.04, (**C**) 25.06, (**D**) 20.06, and (**E**) 30.06 NiTi endodontic rotary files.

None of the NiTi endodontic rotary files (RaCe®, La Chaux-De-Fonds, Switzerland) were discarded after analyzing for possible manufacture defects which could influence the cyclic fatigue resistance of the NiTi endodontic rotary files (RaCe®, La Chaux-De-Fonds, Switzerland). A controlled experimental trial was performed at the Dental Centre of Innovation and Advanced Specialties at Alfonso X El Sabio University (Madrid, Spain), between November and December 2020. The NiTi endodontic rotary files were selected and categorized into the following study groups: A: 250 μm apical diameter and 2% taper (*n* = 10) (25.02); B: 250 μm apical diameter and 4% taper (*n* = 10) (25.04); C: 250 μm apical diameter and 6% taper (*n* = 10) (25.06); D: 200 μm apical diameter and 6% taper (*n* = 10) (20.06); E: 300 μm apical diameter and 6% taper (*n* = 10) (30.06).

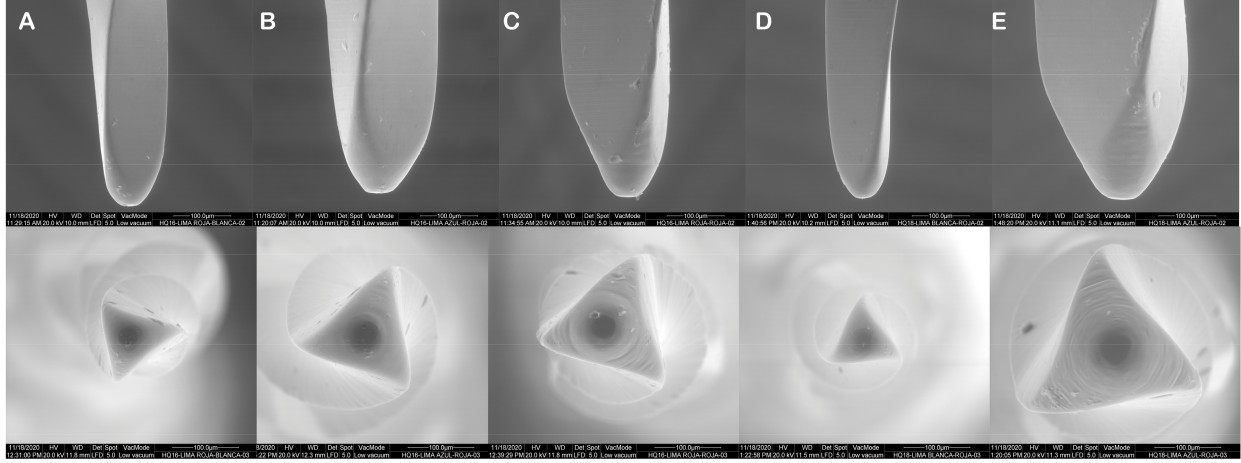

**Figure 2.** SEM analysis of the apical diameter of (**A**) 25.02, (**B**) 25.04, (**C**) 25.06, (**D**) 20.06, and (**E**) 30.06 NiTi endodontic rotary files.

### 2.2. Experimental Cyclic Fatigue Model

The selected NiTi endodontic rotary files (RaCe®, La Chaux-De-Fonds, Switzerland) were used in a custom-made device (utility model patent number ES1219520), designed by computer-aided design/computer-aided engineering (CAD/CAE) 2D/3D software (Midas FX+®, Brunleys, Milton Keynes, UK) and created using 3D printing (ProJet® 6000 3D Systems©, Rock Hill, SC, USA) (Figure 3A–D).

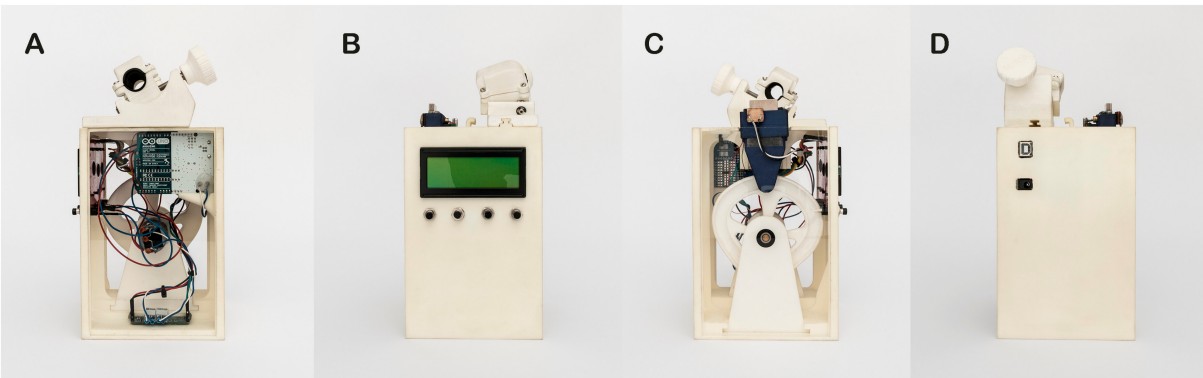

**Figure 3.** (**A**) Front, (**B**) back, and (**C**,**D**) lateral surfaces of the dynamic cyclic fatigue hardware device.

The endodontic rotary file (RaCe, La Chaux-De-Fonds, Switzerland) was submitted to micro computed tomography (Skyscan 1176, Bruker-MicroCT, Kontich, Belgium) with the following exposure parameters: 160.0 kilovolt peak, 56.0–58.0 microamperes, 500.0 ms, 720 projections, four frames, a tungsten target between 0.25 and 0.375 mm, a 3 μm resolution, and a pixel size of 0.127 μm, to obtain a standard tessellation language digital file that allowed the design of an accurate artificial root canal regarding the outer measurements of the selected endodontic rotary file (RaCe, La Chaux-De-Fonds, Switzerland), which ensured intimate contact.

Each one of the previously selected NiTi endodontic rotary files (RaCe, La Chaux-De-Fonds, Switzerland) were introduced in a custom-made artificial root canal designed with a 60° curvature according to Schneider's measuring technique [23] and 3 mm radius of curvature, based on the outer geometry (taper and cross-sections diameters) of each NiTi endodontic rotary file (RaCe, La Chaux-De-Fonds, Switzerland). The artificial root canals were designed using computer aided design/computer aided engineering (CAD/CAE) 2D/3D software (Midas FX+®, Brunleys, Milton Keynes, UK) and manufactured by electri-

cal discharge machining (EDM) molybdenum wire-cut technology (Cocchiola S.A., Buenos Aires, Argentina) from a stainless steel piece 2 mm in width.

The speed of the up-and-down movement of the artificial root canal were generated by the brushed DC gearmotor (Ref.: 1589, Pololu® Corporation, Las Vegas, NV, USA) regarding the signals emitted by the driver (Ref.: DRV8835, Pololu® Corporation, Las Vegas, NV, USA), which performed an H-bridge function that managed the speed of the up-and-down movement through Pulse Width Modulation (PWM) signals emitted by four switches modulated by transistors. The movement generated by the brushed DC gearmotor (Ref.: 1589, Pololu® Corporation, Las Vegas, NV, USA) was transferred to the artificial root canal support through a roller bearing system (Ref.: MR104ZZ, FAG, Schaeffler Herzogenaurach, Germany). The artificial root canal support moved in a pure axial motion through a lineal guide (Ref.: HGH35C 10249-1 001 MA, HIWIN Technologies Corp. Taichung, Taiwan).

The time to failure of the NiTi endodontic rotary files was detected by a light-dependent resistor (LDR) sensor (Ref.: C000025, Arduino LLC®, Ivrea, Italy) located at the apex of the artificial root canal, which received the continuous light source emitted by a high-brightness white light-emitting diode (LED) (20,000 mcd) (Ref.: 12.675/5/b/c/20k, Batuled, Coslada, Spain), which was located opposite to the artificial root canal. The light signals emitted by the LED sensor were detected by the LDR (Ref.: C000025, Arduino LLC®) sensor with a frequency of 50 ms to accurately identify the time of failure. The NiTi endodontic rotary files (RaCe®, La Chaux-De-Fonds, Switzerland) were used with a 6:1 reduction handpiece (X-Smart Plus, Dentsply Maillefer) and torque-controlled motor with continuous rotation at 1000 rpm and 1.5 N/cm torque, according to the manufacturer's instructions [24].

The reduction handpiece (X-Smart Plus, Dentsply Maillefer, Ballaigues, Switzerland) was submitted to an industrial scan (3D Geomagic Capture Wrap, 3D Systems©, Rock Hill, SC, USA) to obtain an STL digital file, which allowed the design (Midas FX+®, Brunleys) and manufacture (ProJet® 6000. 3D Systems©, Rock Hill, SC, USA) of a custom-made support piece placed on the top of the cyclic fatigue testing device to prevent undesirable movements of the reduction handpiece (X-Smart Plus, Dentsply Maillefer) and, hence, the NiTi endodontic rotary files (RaCe®, La Chaux-De-Fonds, Switzerland) inside the artificial root canal. (Figure 4).

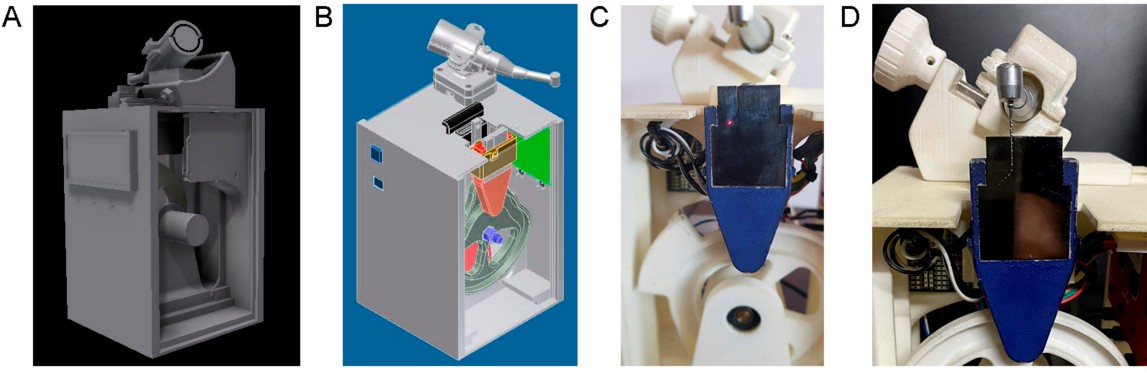

**Figure 4.** (**A**) Back and (**B**) front surfaces of the 3D design of the dynamic cyclic fatigue hardware device, with the reduction handpiece (**C**) located on the top of the dynamic cyclic fatigue hardware device and the NiTi endodontic rotary file (**D**) inside the artificial root canal.

The NiTi endodontic rotary files (RaCe®, La Chaux-De-Fonds, Switzerland) were used until fracture occurred to analyze the effect of the taper and apical diameter on the resistance of NiTi endodontic rotary files (RaCe®, La Chaux-De-Fonds, Switzerland) to cyclic fatigue. All NiTi endodontic rotary files (RaCe®, La Chaux-De-Fonds, Switzerland) were used in the dynamic cyclic fatigue device at a frequency of 60 pecking movements/min, according to a

previous study [25]. To reduce the friction between the reciprocating files and the artificial canal walls, special high-flow synthetic oil designed for the lubrication of mechanical parts (Singer All-Purpose Oil; Singer Corp., Barcelona, Spain) was applied. All NiTi endodontic rotary files (RaCe®, La Chaux-De-Fonds, Switzerland) were used until fracture occurred. The time to failure, the number of cycles to failure, the number of cycles of in-and-out movements, and the length of the fractured file tip were measured and recorded.

### 2.3. Statistical Tests

Statistical analysis of all variables was carried out using SAS 9.4 (SAS Institute Inc., Cary, NC, USA). The descriptive statistics are expressed as the mean and standard deviation (SD) for quantitative variables. Comparative analysis was performed by comparing the time to failure (in seconds), the number of cycles to failure, the number of pecking movements (cycles of in-and-out movements) and the length of the fractured file tip (mm) using the ANOVA test. In addition, the Weibull characteristic strength and Weibull modulus were calculated. The statistical significance was set at $p < 0.05$.

### 3. Results

The mean and standard deviation (SD) values for time to failure (in seconds) regarding the apical diameter and taper are displayed in Table 1 and Figure 5A,B.

**Table 1.** Descriptive statistics of the time to failure in relation to the apical diameter and taper study groups.

| Study Group | $n$ | Mean | SD | Minimum | Maximum |
|---|---|---|---|---|---|
| 20.06 | 10 | 361.91 | 18.07 | 328.18 | 382.91 |
| 25.02 | 10 | 410.39 | 13.11 | 387.28 | 428.19 |
| 25.04 | 10 | 294.27 | 10.50 | 279.87 | 310.33 |
| 25.06 | 10 | 225.77 | 11.73 | 210.79 | 241.83 |
| 30.06 | 10 | 102.91 | 15.41 | 81.29 | 127.91 |

**A**

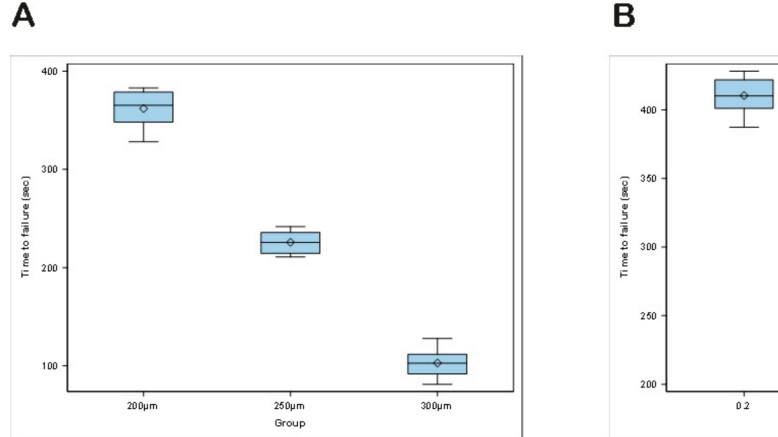

**B**

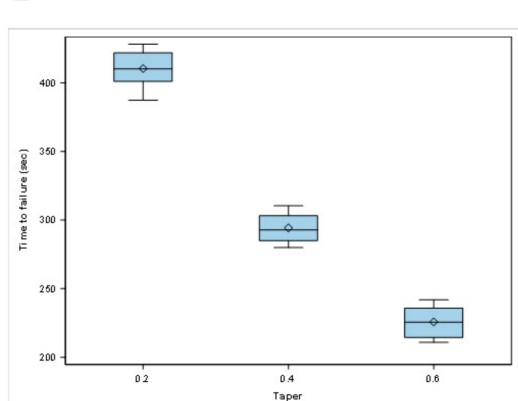

**Figure 5.** Box plots of the time to failure for the (**A**) apical diameter and (**B**) taper study groups. The horizontal line in each box represents the respective median value.

The ANOVA test showed statistically significant differences between the time to failure of the apical diameter ($p < 0.001$) (Figure 5A) and taper ($p < 0.001$) (Figure 5B) of the NiTi endodontic rotary files.

The scale distribution parameter (η) of the Weibull statistics showed statistically significant differences between the time to failure of the 20.06 and 25.06 apical diameter study groups ($p < 0.001$), 20.06 and 30.06 apical diameter study groups ($p < 0.001$) and 25.06 and 30.06 apical diameter study groups ($p < 0.001$). In addition, the shape distribution

parameter (β) of the Weibull statistics showed statistically significant differences between the time to failure of the 20.06 and 25.06 apical diameter study groups ($p < 0.001$) and 20.06 and 30.06 apical diameter study groups ($p = 0.002$); however, it did not show statistically significant differences between the time to failure of the 25.06 and 30.06 apical diameter study groups ($p = 0.656$) (Table 2, Figure 6A).

**Table 2.** The Weibull statistics of the time to failure for the apical diameter and taper study groups.

|  | Weibull Shape (β) | | | | Weibull Scale (η) | | | |
|---|---|---|---|---|---|---|---|---|
|  | Estimate | St Error | Lower | Upper | Estimate | St Error | Lower | Upper |
| 20.06 | 26.22 | 6.75 | 15.83 | 43.43 | 369.77 | 4.70 | 360.67 | 379.09 |
| 25.02 | 37.02 | 9.08 | 22.89 | 59.86 | 416.37 | 3.76 | 409.05 | 423.81 |
| 25.04 | 32.42 | 7.98 | 20.01 | 52.52 | 299.13 | 3.09 | 293.14 | 305.25 |
| 25.06 | 22.37 | 5.53 | 13.78 | 36.31 | 231.14 | 3.46 | 224.45 | 238.02 |
| 30.06 | 7.62 | 1.84 | 4.74 | 12.23 | 109.38 | 4.81 | 100.34 | 119.24 |

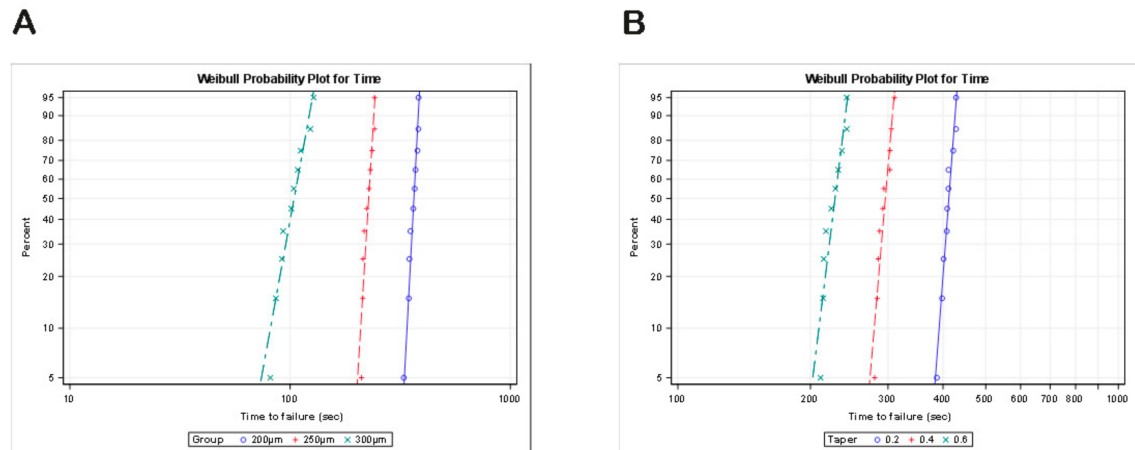

**Figure 6.** Weibull probability plots of the time to failure for the (**A**) apical diameter and (**B**) taper study groups.

The scale distribution parameter (η) of the Weibull statistics showed statistically significant differences between the time to failure of the 25.02 and 25.04 taper study groups ($p < 0.001$), 25.02 and 25.06 taper study groups ($p < 0.001$), and 25.04 and 25.06 taper study groups ($p < 0.001$). However, the shape distribution parameter (β) of the Weibull statistics did not show statistically significant differences between the time to failure of the 25.02 and 25.04 taper study groups ($p = 0.148$), 25.02 and 25.06 taper study groups ($p = 0.287$), and 25.04 and 25.06 taper study groups ($p = 0.702$) (Table 2, Figure 6B).

The mean and SD values for number of cycles to failure regarding the apical diameter and taper are displayed in Table 3 and Figure 7A,B.

**Table 3.** Descriptive statistics of the number of cycles to failure in relation to the apical diameter and taper study groups.

| Study Group | *n* | Mean | SD | Minimum | Maximum |
|---|---|---|---|---|---|
| 20.06 | 10 | 3015.87 | 150.54 | 2734.90 | 3190.80 |
| 25.02 | 10 | 3419.80 | 109.28 | 3227.50 | 3568.30 |
| 25.04 | 10 | 2452.13 | 87.42 | 2332.50 | 2585.80 |
| 25.06 | 10 | 1880.60 | 98.59 | 1749.90 | 2014.90 |
| 30.06 | 10 | 857.57 | 128.32 | 677.40 | 1065.80 |

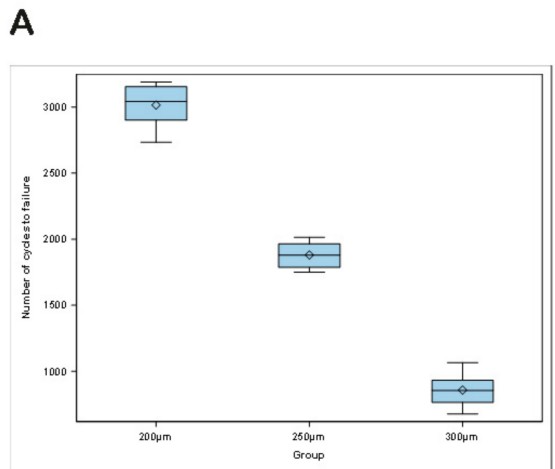

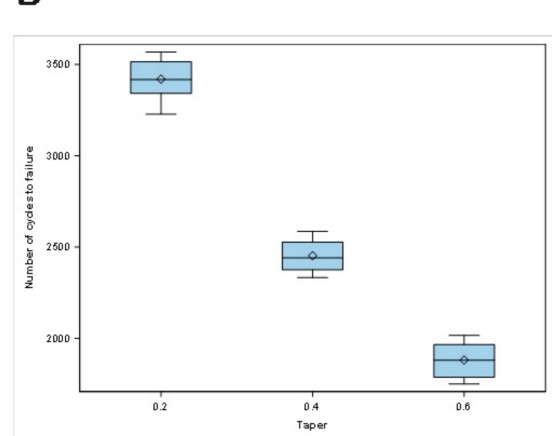

**Figure 7.** Box plots of the number of cycles to failure for the (**A**) apical diameter and (**B**) taper study groups. The horizontal line in each box represents the median value.

The ANOVA test showed statistically significant differences between the number of cycles to failure of the apical diameter ($p < 0.001$) (Figure 7A) and taper ($p < 0.001$) (Figure 7B) of NiTi endodontic rotary files.

The scale distribution parameter ($\eta$) of the Weibull statistics showed statistically significant differences between the number of cycles to failure of the 20.06 and 25.06 apical diameter study groups ($p < 0.001$), 20.06 and 30.06 apical diameter study groups ($p < 0.001$), and 25.06 and 30.06 apical diameter study groups ($p < 0.001$). In addition, the shape distribution parameter ($\beta$) of the Weibull statistics showed statistically significant differences between the time to failure of the 20.06 and 25.06 apical diameter study groups ($p < 0.001$) and 20.06 and 30.06 apical diameter study groups ($p = 0.002$); however, it did not show statistically significant differences between the time to failure of the 25.06 and 30.06 apical diameter study groups ($p = 0.644$) (Table 4, Figure 8A).

**Table 4.** The Weibull statistics of the number of cycles to failure for the apical diameter and taper study groups.

| | Weibull Shape ($\beta$) | | | | Weibull Scale ($\eta$) | | | |
|---|---|---|---|---|---|---|---|---|
| | **Estimate** | **St Error** | **Lower** | **Upper** | **Estimate** | **St Error** | **Lower** | **Upper** |
| 20.06 | 26.24 | 6.76 | 15.84 | 43.47 | 3081.29 | 39.11 | 3005.58 | 3158.91 |
| 25.02 | 37.01 | 9.07 | 22.89 | 59.85 | 3469.64 | 31.38 | 3408.68 | 3531.68 |
| 25.04 | 32.46 | 7.99 | 20.03 | 52.58 | 2492.67 | 25.73 | 2442.74 | 2543.61 |
| 25.06 | 22.25 | 5.51 | 13.07 | 36.14 | 1925.63 | 28.99 | 1869.64 | 1983.29 |
| 30.06 | 7.62 | 1.84 | 4.74 | 12.24 | 911.45 | 40.08 | 836.17 | 993.49 |

The scale distribution parameter ($\eta$) of the Weibull statistics showed statistically significant differences between the number of cycles to failure of 25.02 and 25.04 taper study groups ($p < 0.001$), 25.02 and 25.06 taper study groups ($p < 0.001$), and 25.04 and 25.06 taper study groups ($p < 0.001$). However, the shape distribution parameter ($\beta$) of the Weibull statistics did not show statistically significant differences between the number of cycles to failure of the 25.02 and 25.04 taper study groups ($p = 0.144$), 25.02 and 25.06 taper study groups ($p = 0.279$), and 25.04 and 25.06 taper study groups ($p = 0.705$) (Table 4, Figure 8B).

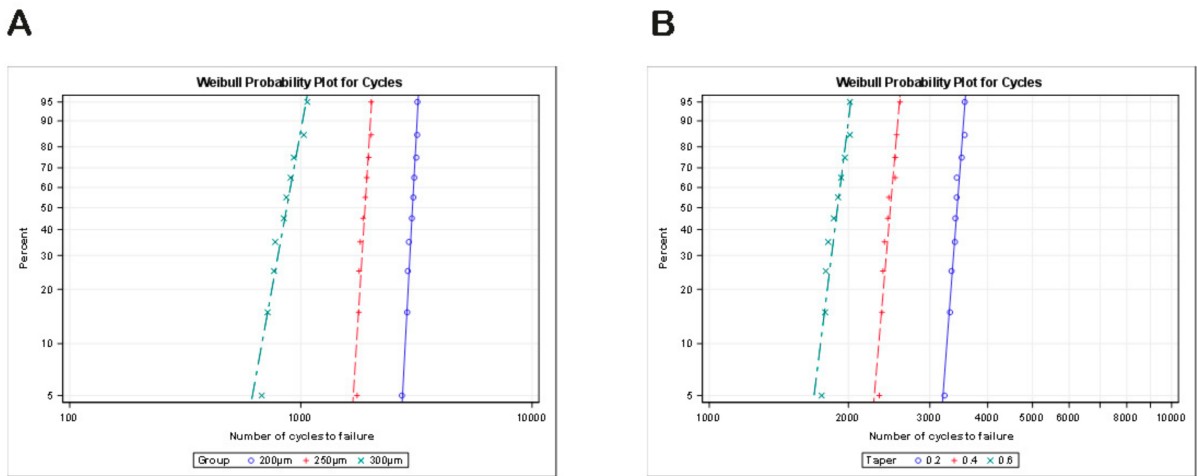

**Figure 8.** The Weibull probability plots of the number of cycles to failure for the (**A**) apical diameter and (**B**) taper study groups.

The mean and SD values for the number of cycles of in-and-out movements regarding the apical diameter and taper are displayed in Table 5 and Figure 9A,B.

**Table 5.** Descriptive statistics of the number of cycles of in-and-out movements in relation to the apical diameter and taper study groups.

| Study Group | n | Mean | SD | Minimum | Maximum |
|---|---|---|---|---|---|
| 20.06 | 10 | 361.91 | 18.07 | 328.18 | 382.91 |
| 25.02 | 10 | 410.39 | 13.11 | 387.28 | 428.19 |
| 25.04 | 10 | 294.27 | 10.50 | 279.87 | 310.33 |
| 25.06 | 10 | 225.77 | 11.73 | 210.79 | 241.83 |
| 30.06 | 10 | 102.91 | 15.41 | 81.29 | 127.91 |

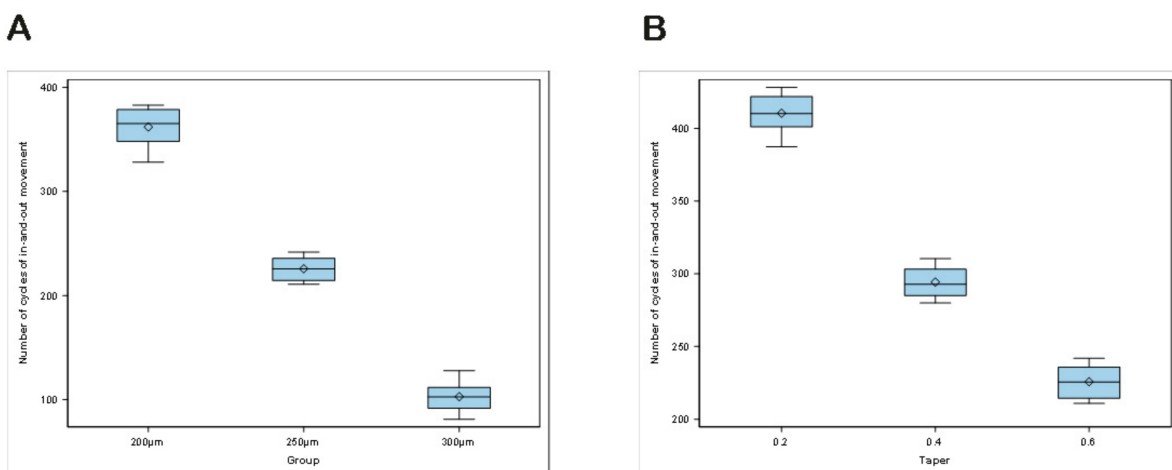

**Figure 9.** Box plots of the number of cycles of in-and-out movements for the (**A**) apical diameter and (**B**) taper study groups. The horizontal line in each box represents the respective median value.

The ANOVA test showed statistically significant differences between the number of cycles of in-and-out movements of the apical diameter ($p < 0.001$) (Figure 9A) and taper ($p < 0.001$) (Figure 9B) of the NiTi endodontic rotary files.

The scale distribution parameter (η) of the Weibull statistics showed statistically significant differences between the number of cycles of in-and-out movements of the 20.06 and 25.06 apical diameter study groups ($p < 0.001$), 20.06 and 30.06 apical diameter study groups ($p < 0.001$), and 25.06 and 30.06 apical diameter study groups ($p < 0.001$). In addition, the shape distribution parameter (β) of the Weibull statistics showed statistically significant differences between the number of cycles of in-and-out movements of the 20.06 and 25.06 apical diameter study groups ($p < 0.001$) and 20.06 and 30.06 apical diameter study groups ($p = 0.002$); however, it did not show statistically significant differences between the number of cycles of in-and-out movements of the 25.06 and 30.06 apical diameter study groups ($p = 0.656$) (Table 6, Figure 10A).

**Table 6.** The Weibull statistics of the number of cycles of in-and-out movements for the apical diameter and taper study groups.

|  | **Weibull Shape (β)** | | | | **Weibull Scale (η)** | | | |
|---|---|---|---|---|---|---|---|---|
|  | Estimate | St Error | Lower | Upper | Estimate | St Error | Lower | Upper |
| 20.06 | 26.22 | 6.75 | 15.83 | 43.43 | 369.77 | 4.70 | 360.67 | 379.09 |
| 25.02 | 37.02 | 9.08 | 22.89 | 59.86 | 416.37 | 3.76 | 409.05 | 423.81 |
| 25.04 | 32.42 | 7.98 | 20.01 | 52.52 | 299.13 | 3.09 | 293.14 | 305.25 |
| 25.06 | 22.37 | 5.53 | 13.78 | 36.31 | 231.14 | 3.46 | 224.45 | 238.02 |
| 30.06 | 7.62 | 1.84 | 4.74 | 12.23 | 109.38 | 4.81 | 100.34 | 119.24 |

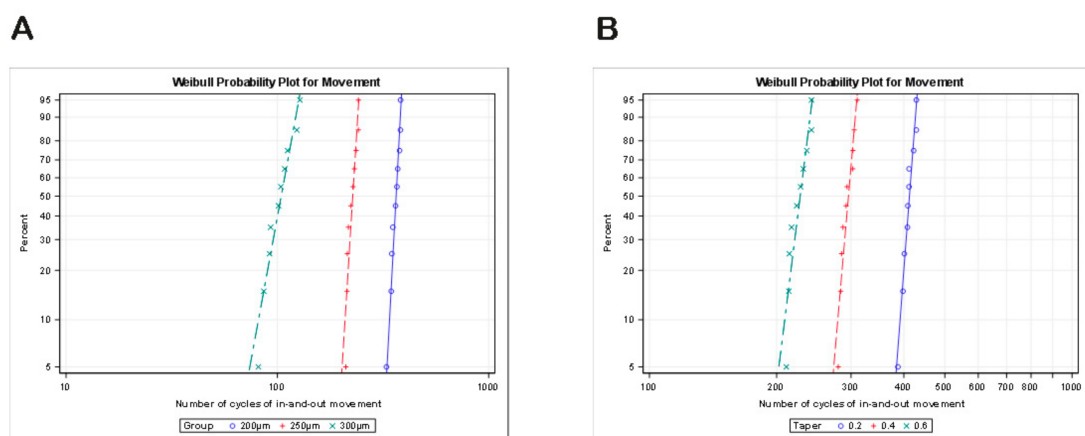

**Figure 10.** Weibull probability plots of the number of cycles of in-and-out movements for the (**A**) apical diameter and (**B**) taper study groups.

The scale distribution parameter (η) and the shape distribution parameter (β) of the Weibull statistics showed statistically significant differences between the number of cycles of in-and-out movements of the 25.02 and 25.04 taper study groups ($p < 0.001$), 25.02 and 25.06 taper study groups ($p < 0.001$), and 25.04 and 25.06 taper study groups ($p < 0.001$) (Table 6, Figure 10B).

The ANOVA test did not show statistically significant differences between the mean length of the fractured files regarding the apical diameter study groups ($p = 0.344$) (Figure 11A) and taper study groups ($p = 0.344$) (Figure 11B).

The scale distribution parameter (η) and the shape distribution parameter (β) of the Weibull statistics did not show statistically significant differences for either the apical diameter ($p > 0.05$) (Figure 12A) or taper ($p > 0.05$) (Figure 12B) study groups.

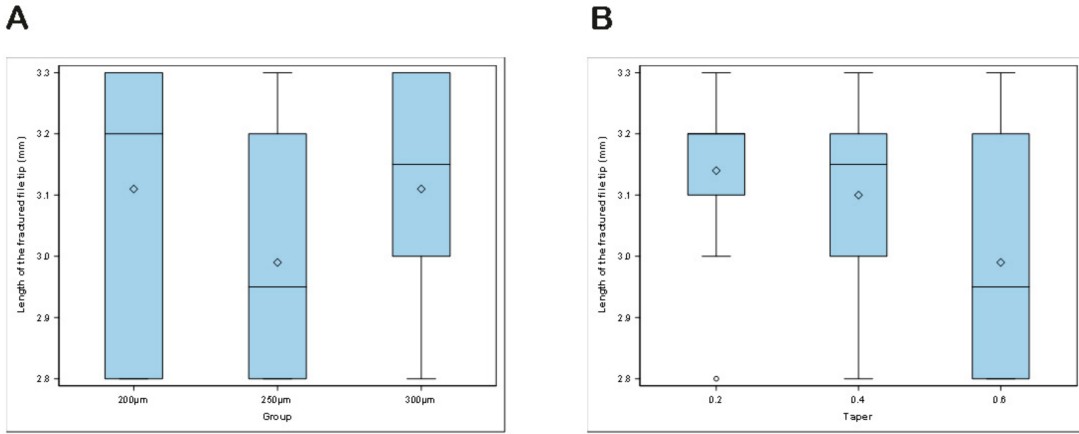

**Figure 11.** Box plots of the length of fractured files regarding the (**A**) apical diameter and (**B**) taper study groups. The horizontal line in each box represents the respective median value.

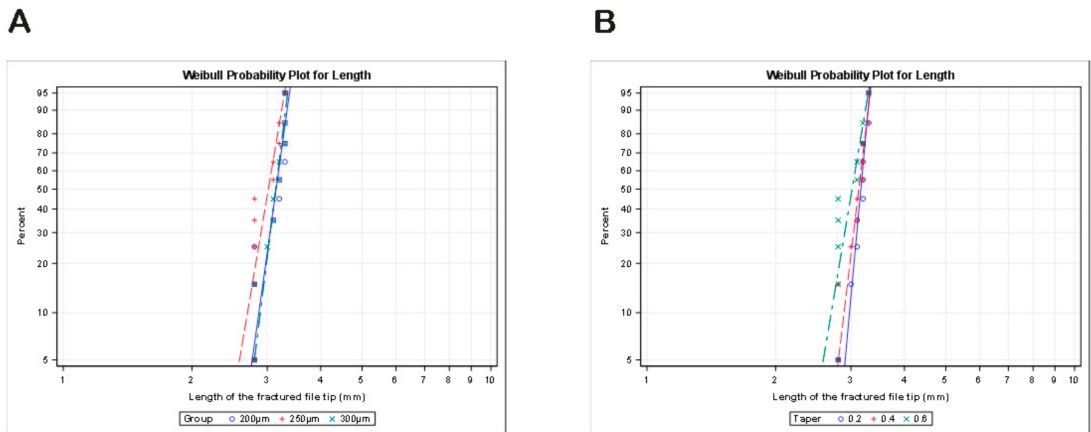

**Figure 12.** The Weibull probability plots of the length of fractured files regarding the (**A**) apical diameter and (**B**) taper study groups.

## 4. Discussion

The results obtained in the present study reject the null hypothesis ($H_0$) which stated that the apical diameter and taper would not affect the resistance of NiTi endodontic rotary files to dynamic cyclic fatigue.

Research reported a prevalence of fractures of NiTi endodontic rotary files between 0.9% [26] and 5% [27], and also highlighted the influence of this intraoperative complication on the outcome of root canal treatment. In addition, Stridberg stated that the fracture of an endodontic instrument inside a root canal system demonstrated a further significant periapical pathology decrease in the success rate of the root canal treatment [28]. Sjögren highlighted the importance of bacterial reduction during the cleaning and shaping procedures of a root canal treatment on the prognosis of the endodontic therapy, and reported that negative microbiological cultures obtained from the root canal system led to an endodontic success rate close to 94%, whereas positive cultures reduced the success rate to 68% [29].

Siqueira pointed out that the persistent bacterial load is the main aetiology factor of endodontic failure and secondary endodontic infections [30]. This is the reason why the cyclic fatigue resistance of NiTi endodontic rotary files has been widely analyzed. Previous studies were conducted to analyze the resistance of both NiTi endodontic rotary and reciprocating instruments to different conditions related to cyclic fatigue [31]. However, the

absence of a normative which regulates the cyclic fatigue tests of the NiTi endodontic rotary and reciprocating files led to the appearance of a heterogeneous multitude of cyclic fatigue test devices, which makes comparison of the results difficult [32]. In the present study, a dynamic cyclic fatigue device was used because it reproduces the operator's movements more accurately, and the results can be extrapolated to the clinical situation [33].

Alcalde et al., showed that the 25.06 NiTi endodontic reciprocating files (ProDesigner R, Easy, Belo Horizonte, Brazil) presented higher ($p > 0.05$) cyclic fatigue resistance and angular rotation before fracture compared to 25.08 Ni Ti endodontic reciprocating files (Reciproc, VDW, Munich, Germany) and 25.07 (WaveOne Gold, Dentsply Sirona, Ballaigues, Switzerland) [34]. However, the different cross-section design, crystalline structure of the NiTi alloy and counterclockwise direction could influence the results obtained in this study.

The NiTi endodontic rotary system used in this study was selected because it provided NiTi endodontic rotary files with different apical diameters while maintaining the cross-sectional design, NiTi alloy crystal structure, and taper, as well as providing NiTi endodontic rotary files with different tapers while maintaining the same apical diameter, cross-sectional design, and crystal structure of the NiTi alloy. In addition, the selection of an NiTi endodontic rotary system instead of a reciprocating system was because the reciprocating movement associated with single-file systems has been shown to extend the lifetime of NiTi endodontic rotary files compared with continuous rotation, thus increasing the cyclic fatigue resistance of the reciprocating files [35].

Even the metallurgical characteristics and thermal treatments of the NiTi alloy influenced the flexibility and cyclic fatigue resistance of NiTi endodontic files [36,37]. For this reason, a conventional NiTi wire alloy endodontic rotary system was also selected. In addition, Gambarini reported the significantly higher cyclic fatigue resistance ($p < 0.01$) of 25.04 NiTi endodontic rotary files (ProFile, Maillefer, Baillagues, Switzerland) compared with 20.06 NiTi endodontic rotary files (ProFile, Maillefer, Baillagues, Switzerland) and 25.06 NiTi endodontic rotary files (ProFile, Maillefer, Baillagues, Switzerland). These results are in line with those of the present study and highlight the influence of the taper above the apical diameter.

Capar et al. also analyzed the cyclic fatigue resistance of rotary pathfinding instruments and reported that the 15.02 NiTi endodontic pathfinding rotary files (HyFlex GPF, Coltene-Whaledent, Allstetten, Switzerland) showed statistically higher cyclic fatigue resistance ($p < 0.05$) compared with the 12.03 NiTi endodontic pathfinding rotary files (G files, Micro-Mega, Besançon Cedex, France), 16.04 NiTi endodontic pathfinding rotary files (ProGlider, Dentsply Maillefer, Ballaigues, Switzerland), 16.02 NiTi endodontic pathfinding rotary files (Pathfile, Dentsply Maillefer, Ballaigues, Switzerland), and 15.02 NiTi endodontic pathfinding rotary files (Scout Race, FKG Dentaire, La Chaux-de-Fonds, Switzerland) [38].

These results agreed with those of the present study, and Camargo et al. reported that the 25.06 NiTi endodontic reciprocating files (ProDesigner R, Easy, Belo Horizonte, Brazil) showed similar canal transportation and centering abilities after preparation of the second mesiobuccal canals compared with the 25.08 NiTi endodontic reciprocating files (Reciproc, VDW, Munich, Germany); however, the 25.06 NiTi endodontic rotary files (ProDesigner R, Easy, Belo Horizonte, Brazil) removed less volume of the root canal dentine and presented an absence of root canal perforation [39].

Duque et al. also reported that the 35.06 NiTi endodontic reciprocating files (WaveOne Gold, Dentsply Sirona, Ballaigues, Switzerland) significantly increased ($p < 0.05$) the percentage of root canal dentine removal compared to the 35.05 NiTi endodontic reciprocating files (ProDesigner R, Easy, Belo Horizonte, Brazil); however, no statistically significant differences ($p > 0.05$) were observed between the apical transportation and percentage of untouched areas of the root canal systems using different apical diameters in curved canals [40].

These results can be summarized in that increasing the mass of the instrument based on an increase in the taper and/or the apical diameter negatively affected the resistance to cyclical fatigue of the instrument; influencing their flexibility and leading the instruments

to cause excessive root canal dentine removal, apical transport, root perforations, and fractures [27,41,42]. The lengths of the fractured files were also measured to determine whether the fracture point would depend on the apical diameter or the taper of the NiTi endodontic rotary instruments, or if it were instead conditioned by the radius of the artificial root canal.

No statistically significant differences were observed between the mean length of the fractured files regarding the apical diameter study groups ($p = 0.344$) and taper study groups ($p = 0.344$); therefore, the length of the fractured files may be associated with the radius of the artificial root canals rather than the apical diameter and taper of the NiTi endodontic rotary instruments. Unfortunately, the limitations of the study prevented the analysis of more tapers and apical diameters and, even with different NiTi alloys, the reciprocating movement and cross-section designs. The study was not developed in a clinical environment due to the difficulty in standardizing the sample.

## 5. Conclusions

The conclusion derived from the present study is that the increase in the apical diameter and taper decreased the cyclic fatigue resistance of NiTi endodontic rotary files; therefore, we recommend enlarging the apical constriction by increasing the apical diameter with low taper instruments.

**Author Contributions:** Conceptualization, V.F.-L.; methodology, N.H.K.; validation, C.R.-S.; formal analysis, I.F.-M.; investigation, V.F.-M. and Á.Z.-M. All authors have read and agreed to the published version of the manuscript.

**Funding:** This research received no external funding.

**Institutional Review Board Statement:** Not applicable.

**Informed Consent Statement:** Not applicable.

**Data Availability Statement:** Data available on request due to restrictions eg privacy or ethical.

**Acknowledgments:** The authors would like to thank Roberto Gutiérrez González and Daniel Ortega Ufano their invaluable assistance in this study.

**Conflicts of Interest:** The authors declare no conflict of interest.

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
