# Peer review of "The Effect of Taper and Apical Diameter on the Cyclic Fatigue Resistance of Rotary Endodontic Files Using an Experimental Electronic Device"

_applsci, doi:10.3390/app11020863_

Round 1

Reviewer 1 Report

Please find some remarks on the paper in the attached file

Author Response

Madrid, December 31, 2020.

Dear Reviewer 1,

I’m pleased to resubmit the manuscript of the work entitled, “Novel Electronic Device to Analyze the Cyclic Fatigue Resistance of Endodontic Rotary Files Regarding the Taper and Apical Diameter”.

Reviewer 1: English language and style are fine/minor spell check required.

Response: In order to respond to the reviewer 1 comment, we have sent the manuscript to the traduction service of MDPI.

Reviewer 1: You begin the title of the paper with Novel Electronic Device … The device is not the main part of your research, but the Analyze of … Novel Elecronic Device should be moved to the end of title, or even removed from title, since in the paper almost nothing told about the device, except some pictures.

Response: In order to respond to the reviewer 1 comment, we have changed the title of the manuscript: “The effect of taper and apical diameter on the cyclic fatigue resistance of rotary endodontic files, using an experimental electronic device”.

Reviewer 1: How were grouped the files in the five sets is irrelevant for the abstract. The mentioned lines might be removed from Abstract.

Response: In order to respond to the reviewer 1 comment, we have removed the aforementioned lines.

Reviewer 1: The most important keyword - Endodontic rotary files – is missing from the Keywords

Response: In order to respond to the reviewer 1 comment, we have added the recommended key word.

Reviewer 1: A study on a similar subject could be referred to in the introduction. Compare the results obtained there with yours (Loios, G.; Martins, Rui F.; Ginjeira, A.; et al. Fatigue resistance of rotary endodontic files submitted to axial motion in multi planar canals manufactured by 3D printing Book Series: Procedia Engineering Volume: 160 Pages: 117-122 Published: 2016

Response: In order to respond to the reviewer 1 comment, we have added the reference in the Introduction section.

Reviewer 1: 50 files were annualized to discard…. Were non-conform parts identified? Please specify that. If yes, were they replaced with some conform ones?

Response: In order to respond to the reviewer 1 comment, we have clarified the sentence.

Reviewer 1: Picture(s) in Figure 3 are not clear enough, and they seem to be irrelevant for the article

Response: In order to respond to the reviewer 1 comment, we consider that the images of the dynamic cyclic fatigue testing device used to fracture the NiTi endodontic rotary files is relevant and helps readers understand the description made in the text.

Reviewer 1: An error message is delivered related to a ?cross reference?: [¡Error! Marcador no definido.].

Response: In order to respond to the reviewer 1 comment, we have clarified the reference.

Reviewer 1: What about the friction before and after lubrication of files? After lubrication of the files the friction between the reciprocating files and the artificial canal walls became lower than the real one? What about the similarity between the experimental and real conditions of work of the files?

Response: In order to respond to the reviewer 1 comment, we clarified that “lubrication is involved in all root canal treatment phases, from dental dam placement to canal obturation. Most often associated with instrumentation, lubrication is required to facilitate the mechanical action of hand/rotary files and to help emulsify and suspend the debris produced” (Chandler N, Chellappa D. Lubrication during root canal treatment. Aust Endod J. 2019 Apr;45(1):106-110.); therefore, we decided to use a lubrication agent to reproduce the clinical situation.

Reviewer 1: It is correct to say that time to failure and length were measured,. It is not correct to say that number of cycles was measured.

Response: In order to respond to the reviewer 1 comment, we clarified that the number of cycles to failure has been used in many studies to analyze the cyclic fatigue resistance of NiTi endodontic rotary files:

Drukteinis S, Peciuliene V, Bendinskaite R, Brukiene V, Maneliene R, Rutkunas V. Shaping and Centering Ability, Cyclic Fatigue Resistance and Fractographic Analysis of Three Thermally Treated NiTi Endodontic Instrument Systems. Materials (Basel). 2020 Dec 21;13(24):E5823.

Alcalde M, Duarte MAH, Amoroso Silva PA, Souza Calefi PH, Silva E, Duque J, Vivan R. Mechanical Properties of ProTaper Gold, EdgeTaper Platinum, Flex Gold and Pro-T Rotary Systems. Eur Endod J. 2020 Dec;5(3):205-211.

Sabet Y, Shahsiah S, Yazdizadeh M, Baghamorady S, Jafarzadeh M. Effect of deep cryogenic treatment on cyclic fatigue resistance of controlled memory wire nickel-titanium rotary instruments. Dent Res J (Isfahan). 2020 Aug 14;17(4):300-305.

Ruiz-Sánchez C, Faus-Llácer V, Faus-Matoses I, Zubizarreta-Macho Á, Sauro S, Faus-Matoses V. The Influence of NiTi Alloy on the Cyclic Fatigue Resistance of Endodontic Files. J Clin Med. 2020 Nov 21;9(11):3755.

Reviewer 1: Figure 3A and B??? Correct is Figure 4 …

Response: In order to respond to the reviewer 1 comment, we have changed the number of the figure.

Reviewer 1: Suggestion: enlarge the pictures in Figure 4, 5, … 11 The text that accompanies the pictures cannot be read.

Response: In order to respond to the reviewer 1 comment, we have increased the size of the aforementioned figures; however, if that is not enough, we will send it to a graphic editor for improvement.

Reviewer 1: Table 3 and Figures 5A and B.??? Correct is Figure 6 ….

Response: In order to respond to the reviewer 1 comment, we have changed the number of the figure.

Reviewer 1: Figure 9. (A) Box plot of the length of fractured files. Correct is Figure 10

Response: In order to respond to the reviewer 1 comment, we have changed the number of the figure.

Reviewer 1: Usually, discussions refer to own work and its output, not to others’ one. Referring to cited papers is used to reveal the other researchers achievements, to emphasize the relevance of the domain approached, and the originality of own work/research. By the way, what do authors claim to be the originality/novelty of the paper?

Response: In order to respond to the reviewer 1 comment, we clarify that the intention of the authors in the discussion section has been to compare and discuss the results obtained in the present study with the findings of other authors. However, differences in study design, instrumentation systems analyzed in other studies and the absence of standardization in the cyclic fatigue testing devices make data comparison difficult. We deeply regret that reviewer 1 has another perception that deviates from the authors' goal when writing the Discussion section.

The novelty of this study is to analyze the influence of taper and apical diameter of NiTi endodontic rotary files on the cyclic fatigue resistance (has never been reported and it is relevant to know how to enlarge the apical diameter of the apical constriction of curved canals) by using a custom-made cyclic fatigue custom device which objectively identifies the time to failure of the NiTi endodontic rotary files every 50ms, the design and manufacture of the artificial root canals allows an intimate adaptation with the NiTi endodontic rotary files that allows contact in its entirety, and allows 10 different speeds of the pecking movement.

Reviewer 1: Conclusions seem to be too hasty presented. How much could apical diameter increase? It is no need a research like the one here presented to conclude that a more robust part has a better resistance against the mechanical stress.

Response: In order to respond to the reviewer 1 comment, we clarify that the authors would have been desired to analyze more tapers and apical diameters to increase the knowledge about how the taper and apical diameter influence on the cyclic fatigue resistance of NiTi endodontic rotary files. Additionally, we clarify that based on the results obtained in this study, both taper and apical diameter variables showed statistically significant differences; therefore, if it is necessary to enlarge the apical diameter of the apical constriction of a curved root canal system, we do not recommend to enlarge with a large diameter and large tapered NiTi endodontic rotary file (such as the 30.06, which had the worst fracture time results: 102.91±15.41 seconds), we recommend to enlarge the apical diameter of the apical constriction of a curved root canal system with a low taper NiTi endodontic rotary instrument (such as 30.02). In resume, based on the results obtained, the increase in the mass of NiTi endodontic rotary instruments; based on the increase in its apical diameter or its taper, leads to a decrease in its flexibility and therefore the cyclic fatigue resistance of NiTi endodontic rotary files to curved root canal systems.

We take this opportunity to thank the recommendations and suggestions made by the reviewers to improve the document.

Yours sincerely,

Reviewer 2 Report

Very interesting study regarding the files materials and anatomy that has a high and important clinical application. Congratulations to the authors

Line 116: There is a typo error

Have the authors considered the limitations of this study? Which are them?

Author Response

Madrid, December 31, 2020.

Dear Reviewer 2,

Reviewer 2: Line 116: There is a typo error

Response: In order to respond to the reviewer 2 comment, we have clarified the reference.

Reviewer 2: Have the authors considered the limitations of this study? Which are them?

Response: In order to respond to the reviewer 2 comment, we have added the study limitations in the Discussion section.

We take this opportunity to thank the recommendations and suggestions made by the reviewers to improve the document.

Yours sincerely,

Reviewer 3 Report

- Title:  The title was slightly confusing, I may for suggest one of the following

           - The effect of taper and apical diameter on the cyclic fatigue resistance of rotary endodontic files, using a novel electronic device.

           - A novel electronic device to analyze the effect of taper and apical diameter on the cyclic fatigue resistance of rotary endodontic files.

Since there is another paper published with the title (Novel Electronic Device to Quantify the Cyclic Fatigue Resistance of Endodontic Reciprocating Files after Using and Sterilization), I am not quite sure if it is still correct to use the word “novel” as to be something new.  May be they can use “experimental”. Im also question why that paper was not used in the references list.

- Abstract:

It seems that the title of the paper emphasized more on the new electronic device (title started with “Novel electronic device”), however, when we look at the abstract and later at the end of the introduction, the device was not mentioned in the aim of the study and it is not clear for me if testing that novel device was part of the aim or no.

Line 16 and 17 : The phrase  nickel-titanium (NiTi) endodontic rotary files was repeated twice I suggest removing one of them .

In line 17: a phrase stated:  "A total of 50 sterile Austenite  wire alloy endodontic rotary instruments" How the author was able to confirm that this is purely Austenite  wire alloy?

Line 26 :  The phrase “movement of both apical diameter” gave me an impression that we are talking about 2 different apical diameters that is why I suggest rephrasing that to “movement of both the apical diameter and the taper” I feel that this phrase is more representative of what is actually being said.

In line 29 a statement mentioned the apical diameter and taper of NiTi endodontic rotary file negatively affect their dynamic cyclic fatigue resistance" this relationship was not clear (like inversely or conversely? or something else)

Introduction:

- Line 34, 35 and 36:  This 1st  phrase in the introduction was referenced to “1”. However, when I checked that reference, I couldnt find any flexibility or elasticity mentioned in that paper which was basically focusing on the cyclic fatigue.  I am not sure if that reference was used properly.

A similar issue was noted with reference 2 and 3, which I do not think a proper citation was done.  It seems to me that there is a secondary citation source used and the author need to refer to the original work which have proven the argument that NiTi files fracture is a problem and that fracture can affect the prognosis of root canal treatment, which I do not think it was the case in reference 2 and 3.

All other references need to be checked for proper citation.

In lines 48 and 49:  Statement that “the influence of taper and apical diameter of NiTi endodontic rotary files on cyclic fatigued resistance has never been reported” is really arguable,  and I will give only 2 references as an example but you can find more if you search the correct database:

     - Plotino, Gianluca, DDS, PhD, Grande, Nicola M., DDS, PhD, Mazza, C., DDS, Petrovic, Renata, MSc, PhD, Testarelli, L., DDS, & Gambarini, Gianluca, MD, DDS. (2010). Influence of size and taper of artificial canals on the trajectory of NiTi rotary instruments in cyclic fatigue studies. Oral Surgery, Oral Medicine, Oral Pathology, Oral Radiology and Endodontics, 109(1), e60-e66. doi:10.1016/j.tripleo.2009.08.009

      - Kingma, C. J. (2014). Influence of taper on the flexibility of nickel-titanium rotary files.

Materials and method:

Study design

- Line 62 mentioned sterile austenite NiTi wire: Since the sterilization can affect the cyclic fatigue resistance I think the author should mention if this is a pre sterile file, or it was processed before use.  

- The other issue is related to the term austenite NiTi wire which is used to differentiate conventional NiTi from other alloys (mechanically or thermomechanically modified alloys like M‐Wire  R‐Phase).  I think the term austenite can be slightly misleading and its better to use “conventional NiTi” instead.

Experimental cyclic fatigue model:

No clear description of the device. The utility model patent number is not enough to describe the device without a reference.  More information is required to clarify the process the device, the set up of the hand-piece, the dynamics of the file movement.

- How tight of a fit these artificial canals were around each file?  Was there any space between the canal wall and the file if yes, how much?

Discussion:

- Between 232 and 235 lines a discussion involve the new improvement in the metallurgic fabrication of a new endodontic files however what the researcher used  here is a conventional or” austenite” files and I do not know how relevant that phrase to his work.

- line 237 : The statement spili etal (reference number 27) mentioned the “incidence of fractured NiTi file”  and numbers were between 0.09% and 5% however that particular paper reported prevalent NOT  incidence and I could not find these numbers in the original paper.

  • There is no clear justification of why this file was chosen I was really confused reading lines between 260 and 267.
  • why 60 picking movement  per/minute? is this is done clinically?

  • How the use of oil lubricant can affect the result and the validity of the research in comparison to reality

There is no clear discussion of this study limitations

The conclusion of the study with regards to enlarging the apical constriction and the use of less taper instrument is not relevant with the study finding. I am not sure why the recommendation was made to enlarge the apical constriction while the study focused on dynamic cyclic fatigue. 

What is the rule of the new cyclic fatigue machine what are the advantages of this machine over other techniques

Author Response

Madrid, December 31, 2020.

Dear Reviewer 3,

I’m pleased to resubmit the manuscript of the work entitled, “Novel Electronic Device to Analyze the Cyclic Fatigue Resistance of Endodontic Rotary Files Regarding the Taper and Apical Diameter”.

Reviewer 3: - Title:The title was slightly confusing, I may for suggest one of the following- The effect of taper and apical diameter on the cyclic fatigue resistance of rotary endodontic files, using a novel electronic device.- A novel electronic device to analyze the effect of taper and apical diameter on the cyclic fatigue resistance of rotary endodontic files.

Response: In order to respond to the reviewer 3 comment, we have changed the title of the manuscript: “The effect of taper and apical diameter on the cyclic fatigue resistance of rotary endodontic files, using an experimental electronic device”.

Reviewer 3: Since there is another paper published with the title (Novel Electronic Device to Quantify the Cyclic Fatigue Resistance of Endodontic Reciprocating Files after Using and Sterilization), I am not quite sure if it is still correct to use the word “novel” as to be something new. May be they can use “experimental”. Im also question why that paper was not used in the references list.

Response: In order to respond to the reviewer 3 comment, we have changed the title and added the reference aforementioned.

Reviewer 3: - Abstract: It seems that the title of the paper emphasized more on the new electronic device (title started with “Novel electronic device”), however, when we look at the abstract and later at the end of the introduction, the device was not mentioned in the aim of the study and it is not clear for me if testing that novel device was part of the aim or no.

Response: In order to respond to the reviewer 3 comment, we clarified that the aim of the study is to analyze the effect of the taper and apical diameter of NiTi endodontic rotary files on the dynamic cyclic fatigue resistance and not the cyclic fatigue testing device. We hope the change of the title will be enough to clarify the aim of the study, without being confusing.

Reviewer 3: Line 16 and 17 : The phrase-nickel-titanium (NiTi) endodontic rotary files was repeated twice I suggest removing one of them

Response: In order to respond to the reviewer 3 comment, we have removed this lines aforementioned.

Reviewer 3: In line 17: a phrase stated: "A total of 50 sterile Austenite wire alloy endodontic rotary instruments" How the author was able to confirm that this is purely Austenite wire alloy?

Response: In order to respond to the reviewer 3 comment, we obtained this information by the manufacturer and it was confirmed in the following article: Aminsobhani M, Khalatbari MS, Meraji N, Ghorbanzadeh A, Sadri E. Evaluation of the Fractured Surface of Five Endodontic Rotary Instruments: A Metallurgical Study. Iran Endod J. 2016 Fall;11(4):286-292.

Reviewer 3: Line 26 : The phrase “movement of both apical diameter” gave me an impression that we are talking about 2 different apical diameters that is why I suggest rephrasing that to “movement of both the apical diameter and the taper” I feel that this phrase is more representative of what is actually being said.

Response: In order to respond to the reviewer 3 comment, we have changed the sentence.

Reviewer 3: In line 29 a statement mentioned the apical diameter and taper of NiTi endodontic rotary file negatively affect their dynamic cyclic fatigue resistance" this relationship was not clear (like inversely or conversely? or something else)

Response: In order to respond to the reviewer 3 comment, we have changed the sentence to make it more clear.

Reviewer 3: Introduction:- Line 34, 35 and 36: This 1st phrase in the introduction was referenced to “1”. However, when I checked that reference, I couldnt find any flexibility or elasticity mentioned in that paper which was basically focusing on the cyclic fatigue. I am not sure if that reference was used properly.

Response: In order to respond to the reviewer 3 comment, we have changed the sentence.

Reviewer 3: A similar issue was noted with reference 2 and 3, which I do not think a proper citation was done. It seems to me that there is a secondary citation source used and the author need to refer to the original work which have proven the argument that NiTi files fracture is a problem and that fracture can affect the prognosis of root canal treatment, which I do not think it was the case in reference 2 and 3.

Response: In order to respond to the reviewer 3 comment, we have replace the references.

Reviewer 3: All other references need to be checked for proper citation.

Response: In order to respond to the reviewer 3 comment, we have replace some references.

Reviewer 3: In lines 48 and 49: Statement that “the influence of taper and apical diameter of NiTi endodontic rotary files on cyclic fatigued resistance has never been reported” is really arguable, and I will give only 2 references as an example but you can find more if you search the correct database:

- Plotino, Gianluca, DDS, PhD, Grande, Nicola M., DDS, PhD, Mazza, C., DDS, Petrovic, Renata, MSc, PhD, Testarelli, L., DDS, & Gambarini, Gianluca, MD, DDS. (2010). Influence of size and taper of artificial canals on the trajectory of NiTi rotary instruments in cyclic fatigue studies. Oral Surgery, Oral Medicine, Oral Pathology, Oral Radiology and Endodontics, 109(1), e60-e66. doi:10.1016/j.tripleo.2009.08.009.

- Kingma, C. J. (2014). Influence of taper on the flexibility of nickel-titanium rotary files.

Response: In order to respond to the reviewer 3 comment, we clarify that the aim of the study conducted by Plotino et al was to “investigate the influence of the shape of 3 different artificial canals on the trajectory followed by different nickel-titanium rotary instruments”. The authors did not analyzed the cyclic fatigue resistance neither fractured the endodontic rotary files. Moreover, the aim of the Thesis conducted by Kingma C.J. was to “determine the influence of taper on the flexibility of various nickeltitanium files”. The author analyzed exclusively the influence of taper but not the apical diameter and the author did not analyzed the cyclic fatigue resistance neither fractured the endodontic rotary files. Therefore, we maintain that “the influence of taper and apical diameter of NiTi endodontic rotary files on cyclic fatigued resistance has never been reported”

Reviewer 3: Materials and method: Study design - Line 62 mentioned sterile austenite NiTi wire: Since the sterilization can affect the cyclic fatigue resistance I think the author should mention if this is a pre sterile file, or it was processed before use.

Response: In order to respond to the reviewer 3 comment, we clarify that the NiTi endodontic rotary files were non-used until they were used in the study. We have clarified this concept in the Abstract section too.

Reviewer 3: - The other issue is related to the term austenite NiTi wire which is used to differentiate conventional NiTi from other alloys (mechanically or thermomechanically modified alloys like M‐Wire R‐Phase). I think the term austenite can be slightly misleading and its better to use “conventional NiTi” instead.

Response: In order to respond to the reviewer 3 comment, we changed the word. We have changed the word in the Abstract and Discussion section too.

Reviewer 3: Experimental cyclic fatigue model: No clear description of the device. The utility model patent number is not enough to describe the device without a reference.  More information is required to clarify the process the device, the set up of the hand-piece, the dynamics of the file movement.

Response: In order to respond to the reviewer 3 comment, we have added two sentences in the Materials and Methods section to clarify the process the device, the set up of the hand-piece and the dynamics of the file movement.

Reviewer 3: - How tight of a fit these artificial canals were around each file?  Was there any space between the canal wall and the file if yes, how much?

Response: In order to respond to the reviewer 3 comment, we have added a sentence in the Materials and Methods section to clarify the procedure of the design and manufacture processes of the artificial root canal.

Reviewer 3: Discussion: - Between 232 and 235 lines a discussion involve the new improvement in the metallurgic fabrication of a new endodontic files however what the researcher used here is a conventional or” austenite” files and I do not know how relevant that phrase to his work.

Response: In order to respond to the reviewer 3 comment, we have removed the sentence.

Reviewer 3: - line 237 : The statement spili etal (reference number 27) mentioned the “incidence of fractured NiTi file” and numbers were between 0.09% and 5% however that particular paper reported prevalent NOT  incidence and I could not find these numbers in the original paper.

Response: In order to respond to the reviewer 3 comment, we have corrected the word (prevalence) and the references where authors found the results.

Reviewer 3: There is no clear justification of why this file was chosen I was really confused reading lines between 260 and 267. Why 60 picking movement per/minute? is this is done clinically?

Response: In order to respond to the reviewer 3 comment, we clarify that the NiTi endodontic rotary system selected in this study provided the authors NiTi endodontic rotary files with the same taper and different apical diameters and same apical diameter and different tapers. This NiTi endodontic rotary system allowed analyzing the influence of “taper” and “apical diameter” variables without variations on the NiTi alloy, manufacturing process and cross section design.

We selected a frequency of 60 pecking movements/min based on the results obtained in a previous study: Zubizarreta-Macho A, Mena Álvarez J, Albadalejo Martínez A, Segura-Egea JJ, Caviedes Brucheli J, Agustín-Panadero R, López Píriz R, Alonso-Ezpeleta O. Influence of the pecking motion on the cyclic fatigue resistance of endodontic rotary files. J. Clin. Med, 2020, 9, pii: E45.

Reviewer 3: How the use of oil lubricant can affect the result and the validity of the research in comparison to reality

Response: In order to respond to the reviewer 3 comment, we clarified that “lubrication is involved in all root canal treatment phases, from dental dam placement to canal obturation. Most often associated with instrumentation, lubrication is required to facilitate the mechanical action of hand/rotary files and to help emulsify and suspend the debris produced” (Chandler N, Chellappa D. Lubrication during root canal treatment. Aust Endod J. 2019 Apr;45(1):106-110.); therefore, we decided to use a lubrication agent to reproduce the clinical situation.

Reviewer 3: There is no clear discussion of this study limitations

Response: In order to respond to the reviewer 3 comment, we have added the study limitations in the Discussion section.

Reviewer 3: The conclusion of the study with regards to enlarging the apical constriction and the use of less taper instrument is not relevant with the study finding. I am not sure why the recommendation was made to enlarge the apical constriction while the study focused on dynamic cyclic fatigue.

Response: In order to respond to the reviewer 3 comment, we clarify that based on the results obtained in this study, both taper and apical diameter variables showed statistically significant differences; therefore, if it is necessary to enlarge the apical diameter of the apical constriction of a curved root canal system, we do not recommend to enlarge with a large diameter and large tapered NiTi endodontic rotary file (such as the 30.06, which had the worst fracture time results: 102.91±15.41 seconds), we recommend to enlarge the apical diameter of the apical constriction of a curved root canal system with a low taper NiTi endodontic rotary instrument (such as 30.02).

Reviewer 3: What is the rule of the new cyclic fatigue machine what are the advantages of this machine over other techniques

Response: In order to respond to the reviewer 3 comment, we clarify that as mentioned in the Materials and Methods section, the cyclic fatigue testing device objectively identifies the time to failure of the NiTi endodontic rotary files every 50ms, the design and manufacture of the artificial root canals allows an intimate adaptation with the NiTi endodontic rotary files that allows contact in its entirety, and allows 10 different speeds of the pecking movement.

We take this opportunity to thank the recommendations and suggestions made by the reviewers to improve the document.

Yours sincerely,

Round 2

Reviewer 1 Report

Corrections applied are correct.

Explanations given to some of the remarks are acceptable

Author Response

Dear Reviewer 1, 

I’m pleased to resubmit the manuscript of the work entitled, “The Effect of Taper and Apical Diameter on the Cyclic Fatigue Resistance of Rotary Endodontic Files, using an Experimental Electronic Device”.

Reviewer 1: I don't feel qualified to judge about the English language and style.

Response: In order to respond to the reviewer 1 comment, we have sent the manuscript to the traduction service of MDPI.

We take this opportunity to thank the recommendations and suggestions made by the reviewer to improve the document.

Yours sincerely,

Reviewer 3 Report

Images that represent the actual device with the handpiece and the rotary file inside the attachment model must be shown. Otherwise reproducibility is in question.

The issue of the length of the fractured piece was not discussed in the discussion. So why it was done?

Clear statement of the limitations is needed.

The conclusion must be rewritten in a different way that reflect the results

Author Response

Dear Reviewer 3,

I’m pleased to resubmit the manuscript of the work entitled, “The Effect of Taper and Apical Diameter on the Cyclic Fatigue Resistance of Rotary Endodontic Files, using an Experimental Electronic Device”.

Reviewer 3: Extensive editing of English language and style required

Response: In order to respond to the reviewer 3 comment, we have sent the manuscript to the traduction service of MDPI.

Reviewer 3: Images that represent the actual device with the handpiece and the rotary file inside the attachment model must be shown. Otherwise reproducibility is in question

Response: In order to respond to the reviewer 3 comment, we have added an image with the handpiece and the rotary file inside the attachment model.

Reviewer 3: The issue of the length of the fractured piece was not discussed in the discussion. So why it was done?

Response: In order to respond to the reviewer 3 comment, we have discuss the issue of the length of the fractured files on the Discussion section.

Reviewer 3: Clear statement of the limitations is needed.

Response: In order to respond to the reviewer 3 comment, we have clarify the limitations of the study.

Reviewer 3: The conclusion must be rewritten in a different way that reflect the results

Response: In order to respond to the reviewer 3 comment, we have rewritten the Conclusion section.

We take this opportunity to thank the recommendations and suggestions made by the reviewer to improve the document.

Yours sincerely,